# Ocular Manifestations in Patients with Philadelphia-Negative Myeloproliferative Neoplasms

**DOI:** 10.3390/cancers12030573

**Published:** 2020-03-02

**Authors:** Charlotte Liisborg, Hans Carl Hasselbalch, Torben Lykke Sørensen

**Affiliations:** 1Department of Ophthalmology, Zealand University Hospital, Vestermarksvej 23, 4000 Roskilde, Denmark; tlso@regionsjaelland.dk; 2Faculty of Health and Medical Sciences, University of Copenhagen, Blegdamsvej 3B, 2200 Copenhagen, Denmark; hkhl@regionsjaelland.dk; 3Department of Haematology, Zealand University Hospital, Vestermarksvej 15-17, 4000 Roskilde, Denmark

**Keywords:** ocular manifestations, ocular complications, philadelphia-negative myeloproliferative neoplasms

## Abstract

The major complications of Philadelphia-negative (Ph-Negative) myeloproliferative neoplasms (MPNs) are thrombosis, haemorrhage and leukemic transformation. As systemic and haematological diseases, MPNs have the potential to affect many tissues and organs. Some complications lead to the diagnosis of MPNs, but other signs and symptoms are often misdiagnosed or neglected as a sign of MPN disease. Therefore, we reviewed the current literature to investigate and delineate the clinical manifestations seen in the eyes of Ph-negative MPN patients. We found that ocular manifestations are common among patients with MPNs. The most frequently described manifestations are due to the consequences of haematological abnormalities causing microvascular disturbances and hyperviscosity. More serious and vision-threatening complications as thrombotic events in the eyes have been repeatedly reported as well. These ocular symptoms may precede more serious extraocular complications. Accordingly, combined ophthalmological and haematological management have the potential to discover these diseases earlier and prevent morbidity and mortality in these patients. Furthermore, routine ophthalmological screening of all newly diagnosed MPN patients may be a preventive approach for early diagnosis and timely treatment of the ocular manifestations.

## 1. Introduction

The Philadelphia-negative (Ph-negative) myeloproliferative neoplasms (MPNs) include essential thrombocytosis (ET), polycythaemia vera (PV) and primary myelofibrosis (PMF), and they are acquired clonal hematopoietic stem-cell disorders. They all share the characteristic that they affect the signal-transduction pathways responsible for haematopoiesis. In addition, the majority of the MPN patients have the JAK2V617F mutation (95–98% of PV patients and in 50% to 60% of ET and PMF patients) [1]. In patients with ET thrombocytosis is prevailing, although a subset of patients may have leukocytosis as well. Patients with PV are characterised by erythrocytosis and—if not present at the time of diagnosis—virtually all patients will develop leukocytosis and thrombocytosis during the course of the disease. The classic PMF is featured by anaemia, variable changes in platelet and leukocyte counts, bone marrow fibrosis, and splenomegaly. The early phase of PMF with thrombocytosis, a normal or only slightly decreased Hb-concentration, and a moderately elevated leukocyte count may mimic ET [2,3]. 

The Ph-negative MPNs as systemic and haematological diseases have the potential to influence many tissues and organs. The major causes of morbidity and mortality are due to the fact that patients are prone to thrombotic and haemorrhagic events and also leukemic transformation [2,3], but other manifestations are seen, including ocular. Ocular signs and symptoms have been described many times in MPN patients, and the complications seen vary and are often secondary to the haematological disturbances pathognomonic for these diseases. These symptoms are often misdiagnosed as other ocular disorders.

To our knowledge, no overview of all the ocular manifestations in these specific patients exists; we, therefore, reviewed the current literature to summarize the ophthalmological complications observed in patients with Ph-negative MPNs. Knowledge of ocular involvement in MPNs is of clinical relevance to these patients and the health personnel in general. The ophthalmologist may play an essential role in patients with MPNs since these patients are often undiagnosed until they develop major thrombotic events. Ocular lesions or symptoms may precede serious extraocular complications related to MPNs. Therefore, the possibility of an underlying haematological disorder should be considered in cases with suspicious or rare ocular findings. Awareness of these symptoms can bring along early recognition and timely initiation of appropriate treatment, which lessens morbidity and mortality in these patients. 

## 2. Method

We searched Pubmed for relevant English language literature. We used the search string “myeloproliferative disorders” [MeSH] AND “Eye diseases” [MeSH]. Moreover, we searched for “Myeloproliferative neoplasms” AND “eye disease” and besides, we made searches with the keywords “ocular”, “ophthalmic”, “eye”, “ophthalmological” along with “myeloproliferative neoplasms”, “complications”, “manifestations”. We did this to find the articles not discovered in the first searches. Finally, we reviewed the references from the gathered publications to eliminate the possibility of overlooking publications not found by the search strategy. We found 49 case stories, 7 case series, 12 studies (observational, interventional or register-based) and two reviews on neurological symptoms in MPNs, including ocular symptoms. We also used references where transient ischemic attacks related to MPNs were described.

## 3. Ocular Manifestations Prevalence and Pathophysiology

Studies suggest that the prevalence of ocular complications in ET and PV patients is high, between 7.5–25% in studies of both treated and untreated patients [4,5,6,7,8,9]. The ocular structures can be involved directly or indirectly. Direct involvement is seen, but the ocular manifestations in MPNs are primarily due to indirect involvement of the ocular structures. These indirect manifestations arise from haematological abnormalities, neurological involvement or may be due to adverse effects from treatment. The most commonly described eye manifestations in ET and PV are symptoms related to the haematological abnormalities of the MPNs such as thrombocytosis, erythrocytosis and leukocytosis. These changes in blood composition give microvascular circulation disturbances caused by aggregation and spontaneous activation of leukocytes and platelets, and hyperviscosity, primarily caused by the elevated haematocrit due to an expanded red blood cell mass. A result of the haematological disturbances is the more severe ocular and vision-threatening manifestations; thrombotic events in the eyes such as arterial or venous occlusion, where vision loss often is irreversible. The microvascular disturbances can also cause inadequate cerebral perfusion and consequently, possible visual symptoms. These vary from monocular blindness, transient blindness, amaurosis fugax, scotomas, hemianopsia, blurring and hallucinations. These neuro-ophthalmological symptoms manifesting in the eyes are often described as visual symptoms related to atypical transient ischemic attacks (TIAs) or migraine-like ischemic attacks (MIAs). The symptoms disappear when elevated blood elements are normalised by treatment [4,5,10,11,12,13,14,15]. 

Finally, other effects or ocular symptoms may occur related to treatment.

It is a different matter with PMF. In the early phase, leukocytosis and thrombocytosis are common and can cause the same symptoms as mentioned above but to a lesser degree [16]. As noted above, the late phase of PMF is characterised by bone marrow failure with anaemia, leukopenia, thrombocytopenia and immunosuppression. Therefore, the manifestations seen in these patients are more often related to anaemia associated with ischemic retinopathy, neovascularisation, retinal haemorrhages and cotton wool spots [17]. Direct involvement of the eyes in the form of extramedullary haematopoiesis is also seen in PMF, similar to different leukaemias [18,19,20,21,22].

The important message for the clinicians is not only to treat the ocular lesions or symptoms according to the standard of care but to consider the possibility of an underlying MPN and initiate timely and appropriate treatment since the symptoms often disappear by treatment of the MPNs. The various ocular manifestations in MPNs are described below and shown in Table 1.

### 3.1. External Eye Involvement 

#### Orbit, Eyelid and Lacrimal Gland Involvement

Involvement of the external structures of the eye is described sparsely in the literature. We found two case reports describing PMF patients, with extramedullary hematopoietic tumours (sclerosing extramedullary hematopoietic tumours, SEMHT) in the orbit and lacrimal gland, respectively [23,24]. Extramedullary haematopoiesis (EMH) is the production of blood cells outside the bone marrow. In most cases, it is the liver, spleen or lymph nodes that are involved, but any organ can be the site of EMH. EMH occurs in many haematological disorders such as leukaemias, lymphomas and haemolytic anaemias and is often the result of ineffective erythropoiesis or bone marrow dysfunction [24,25]. 

We also found two cases of pachydermoperiostosis (PDP) associated with PMF [26,27]. PDP is a disease characterised by soft tissue hyperplasia. Patients with the disease have an appearance of tarsitis with thickened tarsal plates and ptosis. 

### 3.2. Anterior Segment Diseases

#### 3.2.1. Cornea

We found no literature of involvement of the cornea related to MPNs. The cornea is an avascular structure, which is why it seems plausible that the haematological changes in MPNs do not affect this part of the eye.

#### 3.2.2. Conjunctiva

Na et al. described an unusual case report of simultaneous haemorrhage and infarction of the conjunctiva and intraconal space in an ET patient [28]. The risk of haemorrhage increases with high platelet count (>1500 × 10^9^/L), which is thought to be associated with loss of von Willebrand factor resulting in acquired von Willebrand factor syndrome [29]. The haemorrhage made sense in this case, because of the high platelet count (1270 × 10^9^/L), but the authors also found necrotic tissue. Because of the rich blood supply in the conjunctiva, necrosis due to vascular obstruction is uncommon in this structure. Taking into account that the risk of arterial thrombosis is lower with a high platelet count (>1000 × 10^9^/L) [30] it makes the observed infarction unusual.

Another case describes an undiagnosed PV patient with severely injected conjunctivas and unsuccessful treatment for several months where both antibiotics, steroids and artificial tears were attempted. A scraping and conjunctival biopsy were interpreted as vascular congestion and the blood work led to the diagnoses of PV. The eye symptoms disappeared after the initiation of treatment for PV [31].

Extramedullary haematopoiesis is also reported in the conjunctiva for two PMF patients [32].

#### 3.2.3. Iris and Angle

One case of inflammation of the iris, iritis, in an ET patient has been documented [33], and one case of paraneoplastic infiltration of the uveal tract in both eyes resulting in narrow angle-closure glaucoma described [20]. Glaucoma, where the intraocular pressure is high, causing possible damage to the optic nerve, has been reported several times as a cause of underlying MPN disease in all three types of MPNs, and some of the cases where due to neovascular glaucoma secondary to occlusion of the retinal vessels [19,20,34,35].

### 3.3. Posterior Segment Diseases 

#### 3.3.1. Retina and Choroid

The appearance of the fundus in especially untreated or newly diagnosed MPN patients is venous dilation and tortuosity, haemorrhages, cotton wool spots and occasionally Roth spots. These changes often disappear after treatment [14,16,18,19,28,31,33,34,35,36,37,38,39,40,41,42,43], but most of the literature on the subject consists of case stories. A study by Carraro et al. investigated retinopathy and associated fundus lesions in haematological disorders with anaemia and thrombocytopenia [18]. The study included, among other conditions, patients with myelofibrosis and myelodysplastic syndromes. Only three patients with myelofibrosis were included; two of them had fundus lesions. Of the patients with MDS, six of 16 patients had fundus lesions. The fundus lesions discussed in this study were retinal haemorrhages, exudates, papilledema. The authors observed that the prevalence of retinopathy and fundus lesions increased with the severity of the anaemia or thrombocytopenia.

Disease affecting the retina has been reported. Age-related macular degeneration (AMD), is a disease affecting the retina and is the major cause of blindness in developed countries [44]. A fundus photograph of an ET patient with AMD is shown in Figure 1. A large Danish register-based cohort study, including 7958 patients with MPNs and 77,445 controls, found a higher risk of AMD in these patients. The overall adjusted hazard ratio (HR) was reported as 1.3 (95% CI 1.1–1.5). The lowest risk was found for patients with ET (adjusted HR 1.2, 95% CI 1.0–1.6) and the risk increased for patients with PV (adjusted HR 1.4, 95% CI 1.2–1.7) and even further for patients with PMF (adjusted HR 1.7, 95% CI 0.8–4.0). The authors suggest that the association between MPNs and AMD may be explained partly by inflammatory mechanisms [45]. Because of the hyperviscosity and microvascular disturbances in MPN, we speculate that ischemia may also play a part in the development of AMD. Others have shown that reduced choroidal and retinal blood flow is present in patients with both early and late AMD, and therefore ischemia could be a part of the pathogenesis [45,46,47,48]. In accordance, ischemic retinopathy and neovascularisation are documented in PV and PMF patients [49,50], and blood flow in the ophthalmic structures is investigated in a few studies; one by Dapling et al. looked at 15 patients with ET and PV and all but one had normal fluorescein angiograms, suggesting that ongoing microvascular damage is not a significant feature of thrombocytosis due to myeloproliferative disease. Of these 15 patients, four did not receive antiplatelet treatment, and one patient received only cytoreductive treatment in the form of Hydroxyurea. Ten patients received aspirin and five of them in combination with cytoreductive drugs. The platelet counts of the patients were in the range of 393–746 x 10^9^/L. The patients were asymptomatic, and most of them in treatment, and it was concluded that there was no intrinsic retinal vasculopathy in patients with elevated platelet counts secondary to myeloproliferative disease [51].

Yang and colleagues did a similar study where all 374 PV patients with the JAK2 mutation attending their medical centre between 2004 and 2012 were examined. Of those, 53 presented with visual disturbances (including cataract, glaucoma, retinal vascular occlusions, dysfunctional tear syndrome, optic neuropathy and transient ocular blindness). Twenty-one patients who experienced transient blindness episodes underwent fluorescein angiography (FA) before and after treatment. The treatment consisted of phlebotomy to a haematocrit below 50% and Hydroxycarbamide at a dose of 0.5–1.0g daily. The arm–choroidal filling time (ACT) and the artery–venous transit time (AVT) was significantly prolonged in PV patients compared to controls. After treatment, the ocular symptoms disappeared in almost all patients and corresponded with recovered choroidal and retinal blood flow. The patients with persisting symptoms still had elevated platelet counts. Furthermore, the authors of the study found a significant positive correlation between ACT and both haematocrit and platelets, and a significant correlation between AVT and haematocrit, and between AVT and haemoglobin [6]. The results above from the study by Dapling et al. [51] should, therefore, be looked upon with caution because they looked at well-treated/asymptomatic patients and Yang et al observed normalized choroidal and retinal blood flow after treatment.

Willerslev et al. also looked at blood flow in MPN patients. With non-invasive retinal imaging, they demonstrated after successful treatment of the patients increased retinal venous blood velocity, increased retinal arterial blood oxygenation and normalisation of intravascular reflectivity patterns. The study only included seven patients, and further studies are needed to assess the prognostic value of the non-invasive methods they used [39].

Ayintap et al. investigated peripapillary retinal nerve fibre layer thickness (RNFL) in 30 ET patients and compared the thickness with age, sex, race and refractive error-matched controls. ET patient had an average RNFL, which was 8.44% thinner compared to the controls, but the difference was not statistically significant (*P* = 0.226) [52].

Pekel and colleagues investigated subfoveal choroidal thickness (SFCT), retinal vessel calibre, and ocular pulse amplitude between treated PV patients and healthy adults. There were no statistically significant differences between PV patients and controls. They found an association between SFCT and hematocrit level in both PV patients and healthy adults [53]. In a later study, the same research group investigated ET patients, and again choroidal thickness and pulsatile blood flow were not significantly affected in ET patients. They found a statistically significant difference in retinal arteriolar and venular calibres, which are thinner in ET when compared to age and sex-matched healthy controls [54]. To summarize, the only difference between patients with ET and controls in terms of retinal and choroidal structures is the arteriolar and venular calibres.

Another study found a sustained but reversible neuronal hypofunction of the retina by investigating the dark adaptation in 10 PV patients and compared them to 31 healthy controls. They found impaired dark adaptation in the PV patients and the adaptation markedly improved after treatment [55].

Occlusions of the central vessels causing damage to the retina will be addressed in Section 3.4.

#### 3.3.2. Optic Nerve

We found six cases of anterior ischemic optic neuropathy (AION) causing loss of vision due to damage of the optic nerve as a result of ischemia in ET and PV patients, one of them as impending AION and fortunately with full recovery of vision [6,42,56,57,58,59,60,61].

Extramedullary haematopoiesis is described in a case involving the optic nerve sheath [62].

Papilledema is seen after conditions raising the intracranial pressure, for example, after sagittal sinus thrombosis or jugular vein thrombosis described under vaso-occlusions.

### 3.4. Vaso-Occlusions, Haemorrhages and Microangiopathy

Vaso-occlusion is one of the more severe ocular manifestations, where occlusion of central retinal vessels and the cilioretinal artery in most cases cause retinal ischemia and vision loss. Retinal vascular occlusions in MPNs are the result of acquired defects in the blood components and hyperviscosity, as mentioned earlier, causing microvascular disturbances and disruption of the coagulation cascade. Hyperviscosity occurs from either elevation of the cellular or acellular (protein) elements of the blood [6,36,49,63,64,65]. The ophthalmologist must be aware of suspicious ocular thrombotic events, where the patient lacks the typical predispositions for vascular occlusions, such as increasing age, hypertension, diabetes mellitus, smoking and glaucoma. The majority of the MPN patients have the JAK2V617F mutation [1]. The JAK2 mutation in itself induces increased erythrocyte adhesion to the endothelium [66] and the elevated red blood cell count, hematocrit and haemoglobin indicative of PV increase blood hyperviscosity. The same is true for the elevated platelet count in ET patients [40,63,67]. As mentioned earlier, an association between hyperviscosity and vascular disease of the choroid and retina was found by Yang et al. [6].

Both central retinal artery occlusions (CRAO) [35,57,68,69,70,71] and central retinal venous occlusions (CRVO) [34,37,41,72,73] have been reported numerous times in all the three types of MPNs. Other types of retinal occlusions or lateral posterior ciliary artery occlusions have been described [6,14,37,38,72,73,74,75,76,77]. Interestingly a retrospective audit was conducted on JAK2V617F requests to address the clinical value of requesting this mutation status in patients with retinal vein or artery occlusion. Of 17,332 diagnostic requests, 29 included clinical information of either retinal vein/artery occlusion or thrombosis. The haematological abnormalities in these cases were either thrombocytosis, erythrocytosis, elevated haemoglobin and/or haematocrit, or not provided in 10 cases. The JAK2V617F mutation was detected in five patients (17.2%) with either raised haemoglobin and/or haematocrit (n = 3) or thrombocytosis (n = 2) justifying the screening of requesting JAK2V617F mutation status in patients with retinal vein/artery occlusion and an abnormal hemogram [78].

Occlusions in the brain can also cause visual symptoms, and several cases are described. One case of jugular vein thrombosis and following bilateral optic disc oedema with visual loss in a patient with PMF is reported [79] and sagittal sinus thrombosis, some with resulting papilledema are reported in at least six cases associated with PV, and in these cases the ocular manifestation was seen as an initial symptom, [64,80,81].

These cases and studies correlate well with the known fact that these patients have a susceptibility to thromboembolic events.

Haemorrhages in the retina are also seen, and as mentioned in the discussion of the retinal symptoms, the appearance of the fundus in untreated MPNs is often a fundus with venous dilation and tortuosity and haemorrhages [17,18,37,38,51,56,82,83]. Cotton wool spots are occasionally also described in the literature [35,40,41] and Roth spots associated with severe anaemia [20,82].

Other cases involving the vasculature of the retina are chronic vasculitis presenting as temporal arthritis/giant cell arthritis in ET patients [84,85].

A single case of Acute zonal occult outer retinopathy (AZOOR) associated with PV and increased levels of factor VIII has been documented. AZOOR is a syndrome of unknown origin and characterised by rapid loss of outer retinal function in one or more large zones. Because of the increased factor VIII, the authors speculate a thrombotic aetiology of the case of AZOOR [86].

### 3.5. Neuro-Ophthalmologic Symptoms

The microvascular circulation disturbances described earlier give rise to neuro-ophthalmological symptoms. Major cerebrovascular events related to MPN disease, including non-fatal strokes and haemorrhages, can cause ocular symptoms [87,88]. However, minor thrombotic complications such as transient ocular symptoms are the most frequently observed neuro-ophthalmologic symptoms in MPNs. These symptoms are often described as a part of typical or atypical transient ischemic attacks (TIA) and migraine-like ischemic attacks (MIA) and include blurred vision, transient monocular blindness, amaurosis fugax, hemianopsia, scintillating scotomas and hallucinations. Extraocular symptoms of these attacks can be headaches, nausea, vomiting, syncope, seizures and erythromelalgia, which is attacks of burning pain, erythema and warmth of the toes and occasionally the fingers. Numerous studies have reported on this subject, and some of them include visual symptoms. This section is not an exhaustive list of studies and cases reported on the subject, but relevant examples are discussed. We found several case reports, case series and studies [4,5,6,7,8,10,11,12,13,40,65,89,90,91,92,93] reporting on visual symptoms related to TIAs or just reported as “visual disturbances". 

Yang and colleagues retrospectively examined 374 PV patients with the JAK2V617F mutation and 13.6% had visual disturbances as an initial symptom and 41.2% of these cases were presenting as transient ocular blindness [6]. Interestingly, 67% of the patients with transient blindness had previously been misdiagnosed with other ocular disorders. 

Fenaux et al. retrospectively studied the symptoms at diagnosis of 147 patients with ET and found that 7.5% had visual symptoms and 63% of cases were recorded as visual disturbances and the rest as thrombotic events in the ocular structures [7].

Billot et al. described the neurological symptoms that occurred in a series of 37 consecutive ET patients. Neurological symptoms were found in 11 patients (29.7%). Three of the patients had symptoms diagnosed as TIAs and one of those presented with amaurosis fugax. Two other patients had symptoms documented as bilateral visual disturbances. In total, three patients (8.1%) presented with neuro-ophthalmologic symptoms. The authors compared the characteristics of the patients with symptoms to those without and found no difference between the two groups in regard to age, sex, haemoglobin, platelet and leukocyte counts at the time of diagnosis.

Regev et al. recorded symptoms related to ET in 56 patients and followed them for a median of 45 months. Most of the patients received antiplatelet agents (aspirin or/and dipyramidole) and antineoplastic agents (hydroxyurea or busulfan). Visual disturbances were seen in 10.7% and TIA/cerebrovascular accident in 8.9% (no information on related visual symptoms) [8]. The authors of this study found no association between platelet counts and types of symptoms and demonstrated that severe complications are not uncommon in patients with ET despite relatively low platelet counts. 

A lot of studies report on TIA with no information on visual symptoms. The European Collaboration on Low-Dose Aspirin in Polycythemia Vera (ECLAP) project included 1638 patients with PV. They were not newly diagnosed patients and were receiving treatment (phlebotomy, antineoplastic agents or both). At baseline, 19% of the patients had a history of cerebral TIA or stroke. Of the total patients, 1120 were enrolled in a prospective observational study to report on the incidence and risk factors for thrombosis [94]. The study does not report on visual symptoms either at baseline or during the prospective study, illustrating the lack of attention given to ocular symptoms. The other 518 patients from the ECLAP were enrolled in a double-blind, placebo-controlled randomized trial to assess the safety and efficacy of low-dose aspirin treatment in patients with PV [95]. The conclusions were that low-dose aspirin could safely prevent thrombotic complications in patients with PV, including minor thrombotic complications such as TIA. In this randomized trial, it was noted that visual symptoms were a part of TIA, but no estimate of the prevalence of ocular symptoms was given. Numerous other studies have investigated the effects of aspirin on symptoms related to MPNs. Michiels and colleagues’ research is an example of such studies, and they demonstrated that low-dose aspirin reverses the ocular ischemic disturbances and the other complications related to TIAs and MIAs [4,10,11,13,89,96]. 

### 3.6. Ocular Adverse Effects to Treatment

Fraundfelder and Fraundfelder reviewed the literature to find out if interferon alfa therapy was associated with anterior ischemic optic neuropathy (AION). They found 36 case reports of AION described in association with interferon alfa therapy and in 67% of the cases the AION was bilateral. The median duration from treatment start to the onset of AION was 4.5 months. They conclude that the association between interferon alfa treatment and AION can be classified as “possible” according to the World Health Organisation’s causality assessment of suspected adverse drug reactions. The “possible” term of an association is defined as “A clinical event, including laboratory test abnormality, with a reasonable time sequence to the administration of the drug, which could also be explained by concurrent disease, or other drugs or chemicals. Information on drug withdrawal may be lacking or unclear”. Fraundfelder also reports on other ocular events such as blurred vision and irritative conjunctivitis because the drug is secreted in the tears. Finally, ischemic events are reported, but in less than 1% of patients treated and the changes often regress while on the drug or if the treatment is stopped [97]. Another review of Lewczuk et al. suggests a possible association between interferon alpha and retinopathy [98].

Two cases of retinitis in patients treated with Ruxolitinib have been reported. One with cytomegalovirus retinitis due to cytomegalovirus [99] and one with toxoplasmosis due to *Toxoplasma gondii* [100]. Both are cases of opportunistic infections due to the immunosuppressive effect of Ruxolitinib.

## 4. Conclusions

Ocular manifestations are common in patients with MPN and are primarily due to indirect involvement of the ocular structures. The most frequently observed symptoms are those arising from haematological abnormalities due to the elevated blood cell counts and their increased propensity to adhere to each other because of their hyperactivated state, a hallmark of the MPNs. The result is impaired microcirculation consequent to circulating microaggregates of leukocytes and platelets, which typically resolve on treatment with aspirin. 

Inadequate perfusion of retinal vessels increases the risk of devastating vaso-occlusion in the eyes causing loss of vision, with the JAK2V617F mutation in itself adding to the thrombogenic condition by increasing erythrocyte adhesiveness to the endothelium. Insufficient cerebral perfusion causes the most commonly described symptoms, especially in untreated MPN patients, including monocular blindness, transient blindness, amaurosis fugax, scotomas, hemianopsia, blurring and hallucinations. Some of the symptoms are described as part of transient ischemic attacks. 

Other manifestations, such as AMD and hypofunction of the retinal nerve fibre layer, may also be explained by the insufficient blood flow and resulting ischemia, but the inflammatory state in MPN patients may also play a part. 

Ocular manifestations are encountered in any of the MPNs and may precede more serious and potentially life-threatening extraocular complications. Therefore, the health personnel and especially the ophthalmologist should be aware of this entity. The possibility of underlying haematological abnormalities should be considered where an obvious underlying ocular cause or typically predisposing factors are not present. A systemic evaluation and a full haematological workup should be done, so timely recognition of the aetiology of the eye manifestation can be followed by prompt and appropriate treatment. Combined ophthalmological and haematological management has the potential to discover these diseases earlier and prevent morbidity and mortality in MPN patients. Additionally, routine ophthalmological screening of all newly diagnosed MPN patients may be a preventive approach for early diagnosis and timely treatment of the ocular manifestations.

## Figures and Tables

**Figure 1 cancers-12-00573-f001:**
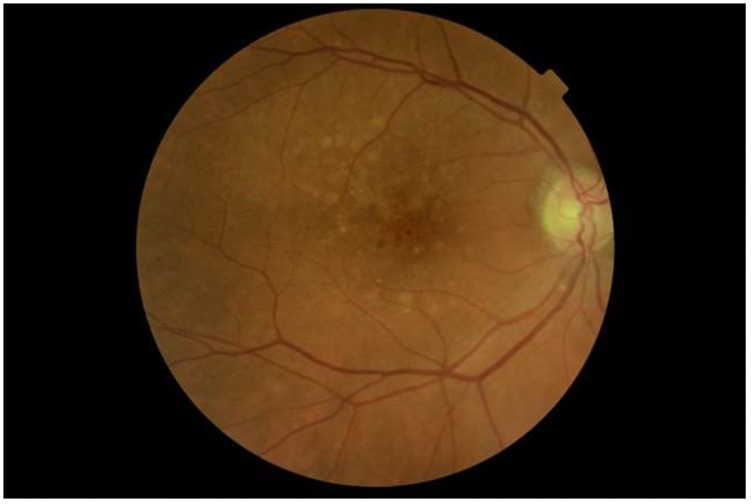
The right eye of a patient with essential thrombocytosis and age-related macular degeneration. Photograph by C.L.

**Table 1 cancers-12-00573-t001:** Ocular manifestations in patients with myeloproliferative neoplasms.

Site or Cause of Involvement	Ocular Manifestations
**External eye involvement**	
Orbit	EMH
Eyelid	Pachydermoperiostosis (PDP)
Lacrimal gland	EMH
**Anterior segment diseases**	
Conjunctiva	Haemorrhage, thrombosis, conjunctivitis/injection
Iris	Iritis
Angle	Neovascular glaucoma
**Posterior segment diseases**	
Retina	Haemorrhages, vascular occlusions, venous dilation and tortuosity, changes in blood flow, cotton wool spots, Roth spots, AMD, peripheral neovascularisation, thinner nerve fibre layer, neuronal hypofunction
Choroid	Changes in blood flow
Optic nerve	AION, EMH, papilledema
**CNS related to eyes**	Blurred vision, transient monocular blindness, amaurosis fugax, scintillating scotomas, hemianopsia, hallucinations
**Secondary to treatment**	
interferon-alpha	Possible AION, retinopathy and irritative conjunctivitis
Ruxolitinib	Opportunistic infections

EMH: extramedullary haematopoiesis, AMD: Age-related macular degeneration, AION: anterior ischemic optic neuropathy. PDP: Pachydermoperiostosis.

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
