# Peer review of "Ocular Manifestations in Patients with Philadelphia-Negative Myeloproliferative Neoplasms"

_cancers, 2020, doi:10.3390/cancers12030573_

Round 1

Reviewer 1 Report

In this article, Liisborg et al. review case reports and studies related to ocular complications in patients with MPN, related to the disease or treatment. There are only few reviews available regarding this subject, which is of importance considering the frequency of such manifestations in this population. The review is quite systematic and all areas of the question are addressed. Overall, this review is of interest to both hematologists who are in charge of MPN patients and ophthalmologists.

A couple of minor points could be improved:

Line 141: "Disease affecting the retina has been reported. Age-related macular degeneration (AMD), is a disease affecting the retina and are the major cause of blindness in developed countries”: improve English

Line 142: a Danish study is mentioned, indicating excess AMD in the MPN population. The authors should mention that the Hazard ratio is relatively weak (HR: 1.3)

Line 216: the study reported (ref41) showed no clear anomaly in fluorescein angiograms and the authors suggest that this is due to the fact that “they looked at well treated patients”. Even though it is true that half patients had cytoreduction (meaning that half did not receive cytoreduction, or just phlebotomies), they kept a high platelet count. These details should be mentioned.

Line 222 : the sentence “Several cases are described: jugular vein thrombosis and following optic disc oedema PMF [77] and sagittal sinus thrombosis are reported in at least 6 cases associated with PV and as an initial symptom, some with resulting papilledema” is unclear. Please reformulate.

Line 229: “the appearance of the fundus in untreated MPNs are especially…” “are” does not seem appropriate

Line 248 : Yang and colleagues who examined 374 PV patients with the JAK2V617F mutation and found 13,6% having visual symptoms and that 41.2% of  these cases were presenting as transient ocular blindness[6]. “who” should be deleted.

Line 275 the statement “The most frequently observed symptoms are those arising from haematological abnormalities due to the elevated blood cell counts and their increased propensity to adhere to each other because of their hyperactivated state” suggests that a link between cell-cell interaction and ocular symptom has been established. I do not believe this to be the case, if so, a reference should be cited.

References: Fraunfelder, F. W.; Fraunfelder, F. T. Interferon Alfa-Associated Anterior Ischemic Optic Neuropathy. 551 Ophthalmology 2011, 118 (2), 402–408. Is wrong, the correct pages are 408-411.

Reviewer 2 Report

In this review, the authors summarized the data available in the literature regarding ocular complications observed in MPN patients. Because these complications are probably underdiagnosed, it seems valuable to gather all studies that reported such manifestations in order to educate clinicians not to miss symptoms that could sign severe complications. However, I have some concerns about the impact of this manuscript if it keeps its current format.

1- In my point of view summarizing the ocular complications observed in MPN patients aims at responding to different questions:

- How frequent are ophthalmological complications?

- How to recognize ocular complications? What are the ocular manifestations that are observed in MPN patients?

- Part 3: What are the pathophysiological mechanisms? How to manage these complications?

This review should not be a list of the different ocular manifestations reported in the literature (with only one or two cases sometimes) as this is currently the case. It should be organized in such a way the most frequent or the most “vision-threatening” complications are exposed. It should also clearly explain how to manage these complications (even if this only represents the management of the MPN itself).

2- Because ocular manifestations are often misdiagnosed, the first question is to expose the frequency of these phenomena that are not rare in MPN patients. This could include the data reported in studies already cited in the manuscript, in particular the study of Yang et al, but also others such as Billot et al. (Haematologica 2011;96(12):1866-1869) or Regev et al (Am J Hematol 1997). It could also be of interest to expose that “large cohort studies” (ECLAP, PT-1 and others) did not reported specifically this type of complications illustrating the lack of attention that hematologist give to ophtalmological symptoms. However, Landolfi et al (NEJM 2005) reported a low incidence of TIA in which visual disturbance where took into account. This kind of informations should be discussed.

3- It could be of interest to search for any association between ocular manifestation and patients characteristics (comorbidities, blood cell counts, etc…). For example, S. Langabeer suggested that a diagnosis of MPN was frequent in retinal vein or artery occlusion when there existed an abnormal hemogram. (Lagabeer, PMID: 30956644). This has not been discussed for most of the ocular manifestations.

4-It could of interest to compare more precisely if there is (or not) any difference between ET and PV patients.

5- Lines 71-73: It should be more explicit that 3 main mechanisms are involved :

- platelet and leukocytes activation and aggregation (probably both in ET and PV)

- ocular infiltration by hematopoietic cells

- hyperviscosity that is more related to PV than ET.

This distinction could be included in a paragraph “pathophysiology” and could serve to differentiate “local ocular complications” (thrombosis, microthrombi in the eyes) from “central causes” linked to cerebral hypoperfusion due to hyperviscosity (and so associated to extra-ocular manifestation such as headaches, …).

6- Lines 84-86: I am not sure that ocular manifestations in PMF can be easily linked to anemia. If this is the case, the authors should compare these complications in MPN patients to patients with similar context with cytopenia such as myelodysplastic syndroms.

7- Lines 121-122: what is the information about this case part that there could be hemorrhagic complications when platelets count is high?

8- Lines 100-128: These paragraphs only represent the description of few cases presenting with MPN and ocular complication. The authors should organize their manuscript in such a way there is clear a message for the clinicians. The case reports should be selected to illustrate more general observations.

9- Lines 138-140: did these observations were correlated with particular symptoms ?

10- Lines 169-180: the data of these different studies could be summarized as follows: the only difference between MPN patients and controls in term of retinal structure is the arteriolar and venular calibres.

11- Lines 170-172: the authors should have conclude that there was not any difference.

12- Lines 190-196 only correspond to a list of cases reports

13- Lines 211-217: as suggested by Yang et al, treatment of MPN is supposed to correct fluorescein angiogram. If the study cited analyzed treated patients, it should not be cited in this review because of an evident bias!

Some minor points could also be addressed :

Lines 14-15: it is not clear why the authors chose to talk about ocular complications. It would be better to explain that MPN can present with ocular complications but that these complications are ofter misdiagnosed, that is why the authors chose to illustrate the different forms of ocular manifestations seen in MPN patients.

Line 19: “disease” should be replaced by “complication”

Lines 28-44: the authors could change the order of the text in the introduction paragraph:

- Introduce the definition of MPN (lines 32-42)

- Then expose the different complication they associated to (28-32)

Line 46 : “to investigate the manifestations seen in the eyes of Ph-negative MPN patients »  could be replaced by « to summarize the ophtalmological complications observed in Ph-negative MPN patients »

Lines 54: I am not sure that a method section is necessary… Moreover it seems that the “methods” are combined with the review of the literature. The second section could could be renamed as “review of the literature”.

In addition, in a lot of studies, visual disturbances are included in neurological complications (probably as a sign of TIA). These terms should have been used to review the data available in the literature.

Line 64: there is no 2.2 paragraph. Therefore a 2.1 paragraph is not mandatory.

Round 2

Reviewer 2 Report

Thank you to the authors for having responded to most of the remarks.